# Peptide fusion improves prime editing efficiency

Minja Velimirovic[1,2], Larissa C. Zanetti[1,3], Max W. Shen[4], James D. Fife [1], Lin Lin[5], Minsun Cha[1], Ersin Akinci [1,6], Danielle Barnum[1,7], Tian Yu[1] & Richard I. Sherwood [1✉]

Prime editing enables search-and-replace genome editing but is limited by low editing efficiency. We present a high-throughput approach, the Peptide Self-Editing sequencing assay (PepSEq), to measure how fusion of 12,000 85-amino acid peptides influences prime editing efficiency. We show that peptide fusion can enhance prime editing, prime-enhancing peptides combine productively, and a top dual peptide-prime editor increases prime editing significantly in multiple cell lines across dozens of target sites. Top prime-enhancing peptides function by increasing translation efficiency and serve as broadly useful tools to improve prime editing efficiency.

---

[1] Division of Genetics, Department of Medicine, Brigham and Women's Hospital and Harvard Medical School, Boston, MA 02115, USA. [2] Centre Hospitalier Universitaire de Québec Research Center–Université Laval, Québec, Québec, QC G1V 4G2, Canada. [3] Hospital Israelita Albert Einstein, São Paulo, SP 05652-900, Brazil. [4] Merkin Institute of Transformative Technologies in Healthcare, Broad Institute of Harvard and MIT, Cambridge, MA 02142, USA. [5] Hubrecht Institute, 3584 CT Utrecht, the Netherlands. [6] Department of Agricultural Biotechnology, Faculty of Agriculture, Akdeniz University, 07070 Antalya, Turkey. [7] Vrije Universiteit Amsterdam, Medical School of V, De Boelelaan 1105, 1081 HV Amsterdam, Netherlands. ✉email: rsherwood@bwh.harvard.edu

Prime editing is a CRISPR-based *'search-and-replace'* technology that mediates targeted insertions, deletions, and all possible base-to-base conversions in mammalian cells in the absence of double-stranded breaks or donor DNA templates[1]. The prime editing enzyme (PE2) consists of SpCas9-nickase fused to an engineered reverse transcriptase (RT). PE2 is recruited to a target site by a prime editing guide RNA (pegRNA) which, in addition to a standard genome-targeting spacer and SpCas9-binding hairpin, contains a 3′ sequence that acts as a template for the fused RT to synthesize a programmed DNA sequence on one of the nicked DNA strands. When cellular DNA repair machinery repairs the broken strand, this RT-extended flap competes with the unedited flap, and the edited sequence sometimes replaces the original sequence in the genome[1,2]. Because of its versatility, prime editing has enormous potential in improving understanding of the effects of genetic changes on cellular and organismal function. However, prime editing is limited by low efficiency. While editing efficiency is dependent on the experimental system, a survey of lentiviral PE2 efficiency at thousands of sites found that PE2 rarely leads to installed edits in >20% of alleles[3]. Analysis of features associated with prime editing efficiency at these thousands of loci found that the strongest correlation is DeepSpCas9 score[3,4], suggesting that prime editing is limited by the interaction strength between the SpCas9-pegRNA complex and the target locus. Optimization of pegRNA features[3], induction of a distal nick on the opposite strand (designated PE3)[1], and pairing overlapping pegRNAs[5] have been found to improve prime editing efficiency, yet low efficiency remains an issue in deployment of prime editing. Here, we present a high-throughput approach, PepSEq, to measure how fusion of peptides influences prime editing and show that peptide fusion enhances prime editing efficiency.

## Results

**Overview of the PepSEq method**. We screened a library of 12,000 85-amino acid peptides derived from DNA repair proteins to identify peptides that improve prime editing efficiency when appended to the N-terminus of PE2. Peptide and protein fusion is a well-established method of modulating genome editing outcomes[6–8]. While scalable, sensitive protein fusion screening remains challenging, high-throughput oligonucleotide library synthesis enables screening of highly diverse peptide fusion constructs. Reasoning that peptides derived from DNA repair-related proteins may encode domains capable of altering prime editing efficiency, we designed a library of 85-amino acid peptides comprising complete 2X tiling of 417 DNA repair-related proteins[9,10] and 29 housekeeping genes as controls (Supplementary Data 1). We also included 5458 DNA repair-related mutant peptides with all possible S– >E and T– >E phosphomimetic substitutions. This library of 12,000 oligos was cloned N-terminal to a 33-amino acid XTEN linker followed by PE2 in a vector allowing Tol2 transposon-mediated genomic integration (Fig. 1a)[11].

To enable quantitative evaluation of peptide-PE2 editing efficiency in high-throughput, we devised the Peptide Self-Editing sequencing assay (PepSEq). We designed a self-targeting pegRNA that introduces a 6 nt mutation (CCTCTG– > GAATTC) in the peptide-adjacent linker sequence (sgPE-linker). Following Tol2-mediated genomic integration of a single peptide-PE2 library member per cell[12], cells are treated in pooled format with sgPE-linker. To evaluate prime editing efficiency in pooled format, we perform paired-end nextgen sequencing (NGS), mapping peptide-PE2 identity and genotypic outcome for each self-targeted allele (Fig. 1a). We performed initial PepSEq screens in mouse embryonic stem cells (mESCs) because they are a non-

transformed cell line not known to possess DNA repair defects, in contrast to other common models such as HEK293T, which is known to lack mismatch repair capacity[13].

We performed three biological replicates of peptide-PE2 integration, each followed by two biological replicates of sgPE-linker addition, collecting >30 M NGS reads for each replicate and 10 M reads prior to sgPE-linker treatment. We developed a computational pipeline to filter and analyze the NGS data (Online Methods), removing peptide-PE2s with <100 total reads in a given replicate from analysis. In total, 16–28% of all alleles were prime edited, 0.3–0.7% were indels, and nearly all remaining reads were unedited (Fig. 1b, c, Supplementary Fig. 1, Supplementary Data 2). Due to the low frequency of indels and other alleles, we focused analysis on prime editing efficiency for each peptide-PE2. Replicates that shared peptide-PE2 integration had strong consistency in prime editing efficiency ($R = 0.42$–$0.66$) while those with distinct peptide-PE2 integration had negligible consistency ($R = 0.01$–$0.06$) (Supplementary Fig. 1). Also of note, wt peptides and their phosphomimetic counterparts did not show significant correlation in relative rates of prime editing efficiency ($R = 0.03$) (Supplementary Fig. 2). It is not surprising to obtain poor replicate consistency in a high-throughput screen in which the majority of library members are expected to be inert, so we analyzed the three combined replicates with independent peptide-PE2 integration using a beta-binomial model to identify 105 top candidate prime editing-enhancing peptide-PE2s (Online Methods).

**Peptide fusions improve PE in multiple mammalian cell lines**. To obtain higher-resolution data on this set of candidate peptide-PE2s, we cloned a sub-library with these 105 peptide-PE2s and 10 control peptide-PE2s, performing PepSEq in five biological replicate peptide-PE2 integrations in mESCs each with >1 M NGS reads. Replicate consistency was much higher ($R = 0.25$–$0.52$, Supplementary Fig. 3, Supplementary Data 3), and the 105 candidate peptide-PE2s as a group gave 15% higher prime editing than control peptide-PE2s ($P < 0.0001$, Fig. 1d), indicating that peptide fusion can improve prime editing efficiency. This screen identified 44 peptide-PE2s that significantly improve prime editing efficiency (FDR = 0.05, Supplementary Data 3), increasing prime editing efficiency up to 70% (Fig. 1e, f). The proteins from which these peptides are derived are not robustly enriched in any particular DNA repair pathway, and none encompass known functional domains that appear related to reported prime editing mechanisms[14]. This result additionally indicates that the 12,000-peptide PepSEq screen was able to flag true hits in spite of noise, a finding supported by the fact that the 44 peptide-PE2s that significantly increase prime editing in the smaller screen have appreciable replicate consistency in the 12,000-peptide screens ($R = 0.17$–$0.39$, Supplementary Fig. 3).

**Peptide fusion effect combines productively**. To gain insight into how these peptides interact, we next asked whether peptides that increase prime editing combine productively. We constructed a dual peptide-PE2 library in which nine top candidate peptides and one control peptide were combined in all 100 possible combinations separated by an eight amino acid linker (Fig. 2a), and we performed ten biological replicate PepSEq screens in mESCs and two replicates in HCT-116 colorectal carcinoma cells. These replicates were highly concordant within and between cell lines (mESC median $R = 0.62$, HCT-116 $R = 0.47$, mESC vs. HCT-116 median $R = 0.32$, Supplementary Fig. 4, Supplementary Data 4), and 79 of the 81 candidate dual peptide-PE2s gave significantly higher prime editing efficiency

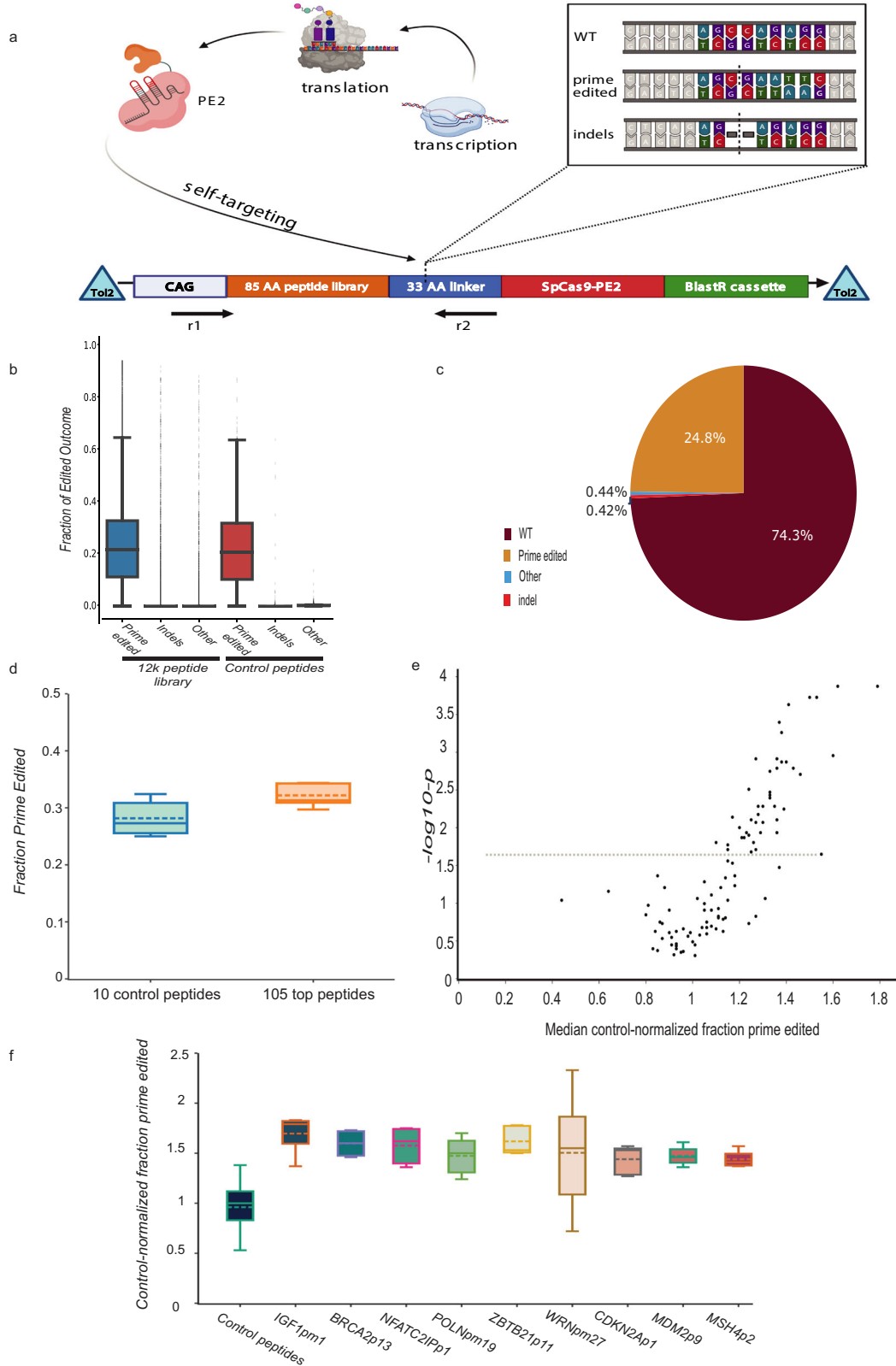

than the control-control pair in mESCs (Fig. 2b, Supplementary Fig. 4).

To explore relationships between peptides, we asked whether dual-peptide-PE2 activation could be predicted by a linear model assuming consistent peptide-specific influences on prime editing. We find high consistency among observed prime editing and expected prime editing under additive assumptions by linear estimates ($r = 0.92$) (Fig. 2c). The high accuracy of the linear model suggests that each peptide has independent (not redundant or synergistic) effects on prime editing, either through interacting with distinct pathways or providing a fixed advantage in protein stability or DNA binding.

Eight of the nine dual peptide-PE2s with highest prime editing included an N-terminal peptide from NFATC2IP

**Fig. 1 The high-throughput Peptide Self-Editing sequencing assay (PepSEq) identifies peptides capable of increasing prime editing efficiency. a** In PepSEq, a library of peptides from human DNA repair-related genes is cloned N-terminal to SpCas9-PE2, separated by a linker, and integrated into cells at one copy per cell. Cells are subsequently treated with a pegRNA targeting the linker sequence that installs a fixed edit. Paired-end genomic DNA NGS of the peptide sequence and the editing site allows calculation of prime editing outcomes in high throughput. **b** Observed prime editing outcome frequencies for a 12,000-peptide PepSEq screen. Box plot indicates median and interquartile range, and whiskers indicate extrema. ($n = 5$ biological replicates). **c** Overall distribution of prime editing outcome frequencies across all 12,000 peptides and all five replicates. **d** Comparison of prime edited allele fraction for 105 DNA repair-related peptides vs. 10 housekeeping control peptides from a 115-peptide PepSEq screen in mESC. Done in four biological replicates ($n = 4$), box plot indicates median and interquartile range, and whiskers go from each quartile to the minimum or maximum. **e** Volcano plot showing control-normalized prime editing fold change (x-axis) vs. vs. $-\log_{10} p$ value (y-axis) from 115-peptide PepSEq screen performed in mESC. **f** Comparison of control-normalized prime edited allele fraction for nine top peptides and all control peptides from a 115-peptide PepSEq screen. Full lines represent median, dash line equals mean of four independent biological replicates. Boxes in (**b**, **d**, **f**) represent the 25–75 percentile ranges with the median of horizontal line. The ends of vertical lines represent minimum or maximum values. The upper and lower whiskers represent scores outside the middle 50%. ns not significant by the Paired Student's two-tailed $t$ tests were performed to calculate $p$ values. Source data and exact $p$ values are provided as a Source Data file. Statistically significant differences are denoted as follows: $*p < 0.05$, $**p < 0.01$, $***p < 0.001$.

(NFATC2IPp1), and the dual-peptide with highest median prime editing in mESC and HCT-116 (median 1.77X control-control in mESC, 1.88X in HCT-116) pairs NFATC2IPp1 with a phospho-mimetic peptide from IGF1 (IGF1pm1) (Fig. 2d). These two peptides induce the strongest increases in prime editing in the single-peptide-PE2 screen (Fig. 1c) and in the linear model of dual-peptide-PE2 screen (Supplementary Data 4), providing rationale to pursue IGFpm1-NFATC2IPp1-PE2 (IN-PE2) as an improved prime editor.

**Effects of editing type and position on IN-PE2 efficiency**. To ask whether IN-PE2 increases prime editing efficiency across a larger collection of target sites, we designed a lentiviral library comprising 100 pegRNA-target pairs spanning a range of edit types and predicted editing efficiencies[3]. After stable integration of this library in three human and mouse cell lines (mESC, HEK293T, U2OS), we treated cells with either IN-PE2 or a control PE2 containing the 5′ linker sequence but lacking an N-terminal peptide (CTRL-PE2). Among the targets with sufficient library representation and editing, IN-PE2 yielded significantly higher prime editing than CTRL-PE2 in all three cell lines (median 1.63X in mESC at 19 sites, 1.31X in HEK293T at 27 sites, 1.23X in U2OS at 12 sites) Fig. 2e, Supplementary Fig. 5, Supplementary Data 5). The results indicate that IN-PE2 leads to a consistent increase in prime editing efficiency across a variety of targets (Supplementary Fig. 5).

We performed pegRNA-target library experiments with six additional peptide-PE2s with one to three top candidate peptides fused to PE2, finding IN-PE2 to display the most robust prime editing of these seven peptide-PE2s (Supplementary Fig. 5). For each peptide combination, changes in prime editing efficiency are largely consistent across the 13 sites with >0.5% editing efficiency (Supplementary Fig. 6), suggesting that PE2-enhancing peptides are not highly site-specific. We note that the tested tri-peptide PE2 fusions consistently have lower prime editing efficiency than other PE2 fusions, which may indicate that beyond di-peptide fusions, the advantage of adding prime-enhancing peptides is offset by increased steric hindrance or diminished translation efficiency. The consistent increase in prime editing efficiency in cell lines without known DNA repair defects (mESC, U2OS) and those with known deficiencies in mismatch repair (HEK293T[13], HCT-116[15]) suggests that IN-PE2 is unlikely to function through interaction with mismatch repair machinery.

**IN-PE2 improves PE of therapeutically relevant mutations**. We next asked whether IN-PE2 increases prime editing at endogenous genomic loci. We designed 12 pegRNAs to install missense variants (6 in NF2 and 6 in TP53) and evaluated prime editing

efficiency in 4 cell lines (HEK293T, U2OS, A549, and HCT-116). We find that, across all targeted loci, IN-PE2 treatment yields significantly increased prime editing efficiency in all cell lines as compared to PE2 (median 1.35X in A549, 1.92X in HCT-116, 1.17X in HEK293T, 1.78X in U2OS, $p < 0.01$ in each cell line) (Fig. 2f, Supplementary Figs. 7 and 8). While there is variability in the increase in prime editing at individual sites and across cell lines, these results indicate that IN-PE2 on average provides an increase in prime editing at endogenous loci.

**Basis of enhanced PE with IN-PE2**. We next investigated mechanisms by which prime-enhancing peptides function. NFATC2IPp1 (AA2-86) is in a disordered region with no prior annotated function. IGFpm1 (AA74-154) spans regions involved in IGF1 interaction with insulin receptor and IGF receptors, as well as a disordered region. As these IGF1 interactions occur extracellularly, protein annotations do not clarify how these peptides function to enhance prime editing. Moreover, the nuclear localization signal (NLS) algorithm cNLS mapper[16] does not predict either peptide to harbor an NLS.

We reasoned that IN-PE2 may increase prime editing efficiency through increased cellular expression. To evaluate this idea, we constructed IN-GFP-PE2 and CTRL-GFP-PE2 fusions. We found that mESCs possess 1.58X the amount of IN-GFP-PE2 as CTRL-GFP-PE2 ($N = 5$, $p < 0.0001$, Fig. 2g, Supplementary Fig. 9, Supplementary Data 8) due to a population-wide shift in GFP levels. Cycloheximide time course experiments show that IN-GFP-PE2 and CTRL-GFP-PE2 are degraded at a similar rate (Supplementary Fig. 9), altogether suggesting that the IN peptides increase either transcription or translation of the PE2 enzyme and offering a plausible explanation for the increased activity of IN-PE2. In order to demonstrate a link between increased translation/transcription and prime editing stimulation, we sorted cells constitutively expressing IN-GFP-PE2 and dosed with a self-editing pegRNA into three populations based on GFP fluorescence (top 20%, next 20% and bottom 60%). We found a positive correlation between intensity of GFP fluorescence and prime editing rates, indicating that higher levels of PE2 lead to increased peptide-enhanced prime editing ($N = 2$, $p = 0.007$, Supplementary Fig. 9d). We also find that this increase in protein levels is not unique to IN-GFP-PE2, as mESCs expressing IN-GFP possess 2× the amount of GFP expression as cells expressing a CTRL-GFP construct ($N = 3$, $p = 0.008$, Supplementary Fig. 10).

To distinguish between increased transcription and translation, we compared transcript levels of IN-GFP and CTRL-GFP in mESCs by RT-qPCR, finding that CTRL-GFP is expressed at 1.3-fold higher levels as IN-GFP ($N = 3$, $p = 0.0006$, Supplementary Fig. 10), even as cells possess higher levels on IN-GFP protein. Finally, if the IN peptides act to increase translation efficiency, we

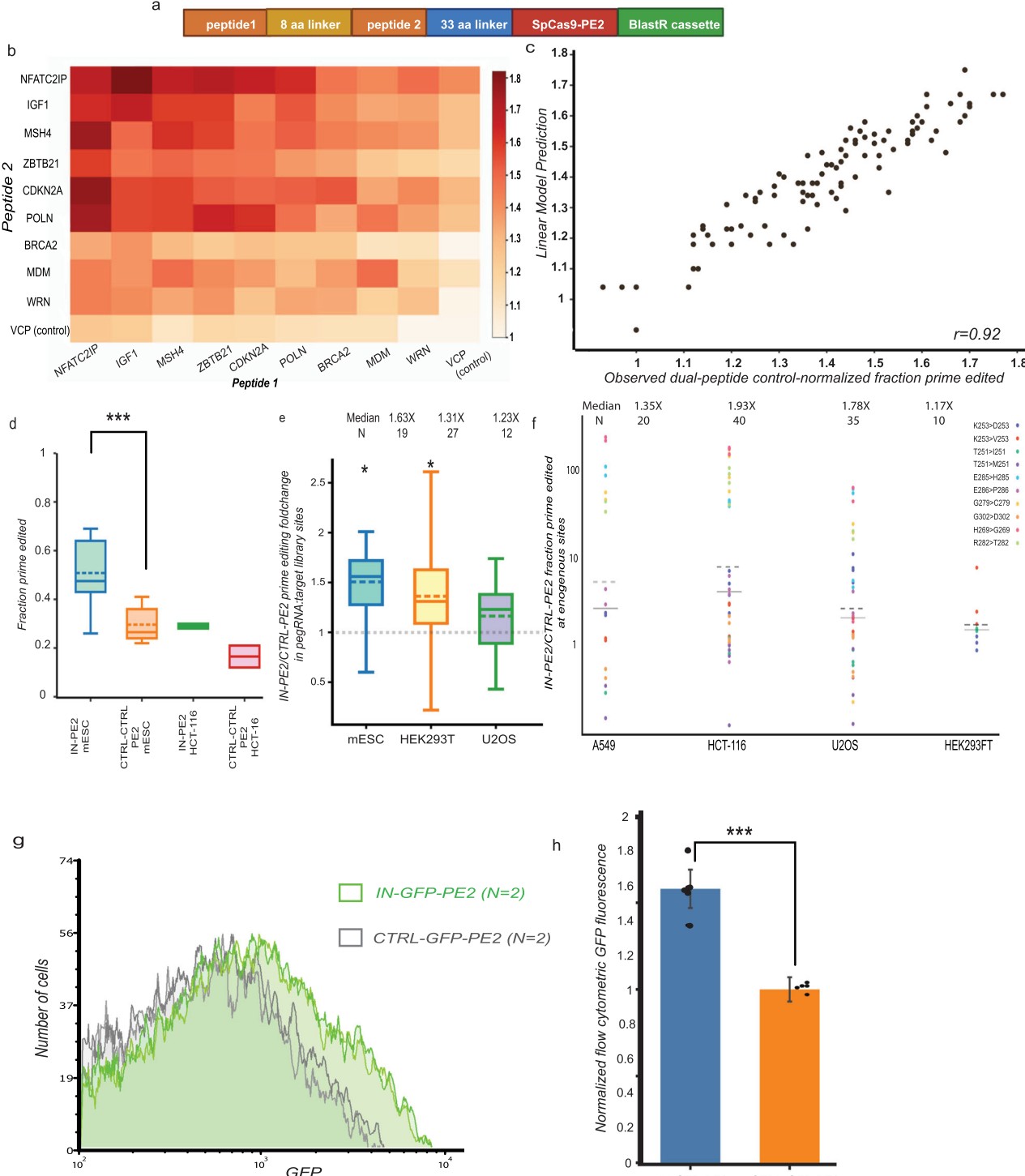

**Fig. 2 A dual-peptide-PE2 displays improved prime editing efficiency across dozens of loci in four cell lines. a** Construct design for a dual-peptide PepSEq library with all possible pairs of top 10 peptides. **b** Comparison of control-normalized prime edited fraction for 81 dual-peptide pairs. **c** Comparison of control-normalized dual-peptide prime edited fraction predicted by a linear model vs. observed median prime edited fraction. **d** Comparison of prime edited fraction for IN-PE2 vs. control-control-PE2 in mESC ($n = 8$ independent replicates) and HCT-116 ($n = 4$ replicates) dual-peptide screen. Bars represent mean signal (solid line) and median (dotted line); **e** Comparison of median control-normalized prime edited fraction for IN-PE2 in the 100-target library across three cell lines. Bars represent mean signal (solid line) and median (dotted line); $n = 19$; 27; 12 independent replicates. **f** Comparison of prime edited fraction for IN-PE2 vs. CTRL-PE2 at a set of twelve endogenous sites in HEK293T, HCT116, A549 and U2OS. $n = 4$ independent replicates. **g** Flow cytometric GFP fluorescence intensity for two representative replicates of IN-GFP-PE2 vs. CTRL-GFP-PE2 in mESCs. **h** Comparison of IN-GFP-PE2 vs. CTRL-GFP-PE2 control-normalized flow cytometric GFP fluorescence levels. $n = 5$ biological replicates. Boxes in (**d**, **e**) represent the 25–75 percentile ranges with the median of horizontal line. The ends of vertical lines represent minimum or maximum values. ns not significant by the Paired Student's two-tailed $t$ tests were performed to calculate $p$ values. Error bars in all figures represent SD. Source data and exact $p$ values are provided as a Source Data file. Statistically significant differences are denoted as follows: $*p < 0.05$, $**p < 0.01$, $***p < 0.001$.

hypothesized that they should increase prime editing efficiency even when decoupled from PE2. Therefore, we cloned a construct in which the IN peptides are separated from PE2 by a self-cleaving 2 A peptide (IN-P2A-PE2). Using the PepSEq assay in mESCs, we found that IN-P2A-PE2 leads to significantly improved prime editing as compared to CTRL-PE2, editing with comparable efficiency to IN-PE2 (median 1.87×, $p = 0.005$, Supplementary Fig. 10, $N = 3$). Altogether, we posit that the IN peptides work by increasing translation, and the resulting increased levels of PE2 in the cell increase prime editing efficiency. How these peptides act to increase translation remains an open question, and future use of PepSEq might employ scanning mutagenesis to map residues that are crucial to function of the IN peptides.

## Discussion

In summary, through screening 12,000 peptide-PE2 fusion proteins using PepSEq, a sensitive, NGS-based self-editing platform, we identify a prime editor that consistently increases editing efficiency across dozens of targets in four human and mouse cell lines. There have been a number of orthogonal approaches that have recently been shown to increase prime editing efficiency[5,14,17–19]. Prime editing has been shown to be inhibited by the MMR pathway and improved by co-expression of a dominant negative MLH1 isoform[14,17]. However, this improvement is only observed for certain types of edits, presumably due to differential recognition by distinct MMR subcomplexes. Addition of stabilizing RNA structures to the pegRNA in the form of structured RNA motifs (epegRNAs) or circularization has also been found to increase prime editing efficiency[18,19]. We note that prime editing shows variability across endogenous target sites in the presence and absence of IN-PE2, as is the case with other improved prime editing tools. It will be fruitful to establish whether these approaches combine with the IN dipeptide to further increase the efficacy and thus utility of prime editors.

Prime editing applications have also been expanding rapidly. TwinPE, a method that uses a prime editor protein and two pegRNAs for excision of DNA sequences at endogenous human genomic loci, expands the capabilities of prime editing to include targeted deletion[20]. When combined with site-specific recombinases, twinPE can be used for gene-sized integration without requiring double-strand breaks, expanding the range of genome editing capabilities. Prime editing has also been used to correct disease models in mice[21,22]. We anticipate that IN-PE2 will improve the utility of prime editing for these diverse applications.

## Methods

**Peptide library design**. We designed 85-amino acid peptides covering all annotated human DNA repair proteins[23,24], tiling by starting each peptide 45-amino acids after the prior peptide using a codon-optimized library design[25]. We also included mutant peptides with all possible S– > E and T– > E phosphomimetic substitutions. 147 wild-type peptides targeting 29 housekeeping genes were also included as controls. Unique 9 nt sequences were inserted in phosphomimetic peptides to facilitate sequence mapping for downstream analysis. The sequence design was performed with "seqinr" and "Biostrings" packages in R.

**Cell culture**. All cell lines were obtained from ATCC and were cultured in: McCoy's 5 A media (Thermo Fisher) + 10% FBS (Thermo Fisher) (U2OS, HCT-116); DMEM (Thermo Fisher) + 10% FBS (HEK293); mESCs were maintained on gelatin-coated plates feeder-free in mESC media composed of Knockout DMEM (Life Technologies) supplemented with 15% defined foetal bovine serum (FBS) (HyClone), 0.1 mM nonessential amino acids (Life Technologies), Glutamax (Life Technologies), 0.55 mM 2 -mercaptoethanol (b -ME) (Sigma), 1X ESGRO LIF (Millipore), 5 nM GSK-3 inhibitor XV and 500 nM UO126. Cells were regularly tested for mycoplasma. HEK293T (ATCC CRL-3216), U2OS (ATCC HTB-96), HCT 116 (ATCC CCL-247).

**Peptide library cloning and screening**. The SpCas9-PE2-encoding sequence from pCMV-PE2[1] (Addgene Plasmid #132775) was subcloned into the p2T-CAG-SpCas9-BlastR plasmid[26] (Addgene Plasmid #107190) to create p2T-CAG-SpCas9PE2-5pLinker-BlastR (Addgene Plasmid #173066). This plasmid was later used to clone in peptides, such as IN dipeptide making p2T_CAG_SpCas9PE2_Igf1-NFATC2IP_BlastR (Addgene Plasmid # 173067).

Specified oligonucleotide libraries were synthesized by Twist Bioscience (12,000-peptide) or IDT (115-peptide and dual-peptide) and were cloned into the NheI site of p2T-CAG-SpCas9PE2-5pLinker-BlastR through amplification with Q5® High-Fidelity DNA Polymerase (New England Biolabs) using primers Cas9NTLib_GA_fw and Cas9NTLib_GA_rv (see Supplementary Data 6) followed by ligation using the NEBuilder HiFi DNA Assembly Kit (NEB) for 1 h at 50 °C. Assembled plasmids were purified by isopropanol precipitation with GlycoBlue Coprecipitant (Thermo Fisher) and reconstituted in TE and transformed into NEB® 10-beta Electrocompetent *E. coli* (NEB). Following recovery, the library was grown in liquid culture in LB medium overnight at 37 °C and isolated by ZymoPURE™ II Plasmid Maxiprep Kit (Zymo Research). Library integrity was verified by restriction digest with AgeI (New England Biolabs) for 1 h at 37 °C, and library diversity was validated by Sanger sequencing sampling.

Mouse ESC cells were plated at ~20–25% confluence onto 25 cm plates the day before transfection so that they reach ~50–75% confluency on the day of transfection. For stable Tol2 transposon plasmid integration, cells were transfected using Lipofectamine 3000 (Thermo Fisher) following standard protocols, and equimolar amounts of Tol2 transposase plasmid and transposon-containing plasmid. To generate lines with stable Tol2-mediated genomic integration, selection with the appropriate selection agent at an empirically defined concentration (blasticidin, hygromycin, or puromycin) was performed starting 24 h after transfection and continuing for >1 week.

In cases where sequential plasmid integration was performed such as integrating pegRNA/target library and then Cas9, the same Lipofectamine 3000 transfection protocol with Tol2 transposase plasmid was performed each time, and >1 week of appropriate drug selection was performed after each transfection.

**Deep sequencing, library preparation**. Genomic DNA was extracted from harvested cells with the PureLink Genomic DNA Purification Mini Kit (Invitrogen). For library experiments, sequences including the peptide and the prime editing site were PCR amplified using Q5® High-Fidelity DNA Polymerase (NEB) and primers as specified (Supplementary Data 6). For each replicate, the first PCR included a total of 10–20 µg of genomic DNA. To determine the number of cycles required to complete the exponential phase of amplification we first performed qPCR, followed by PCR using primers that included both Illumina adaptor and barcode sequences (Supplementary Data 6). For measuring PE2 efficiencies at endogenous sites, the independent first PCR was performed in a 200 ul reaction volume that contained 1000 ng of the initial genomic DNA template per sample. The second PCR to attach the Illumina adaptor and barcode sequences was then performed using purified product from the first PCR. After bead purification, pooled samples were sequenced using NextSeq (Illumina).

**Library data processing**. Designed library peptides were identified in sequenced reads by exact string matching to the first eight nucleotides of the peptide sequence, which were unique across the library. Sequenced target sites were aligned to the designed reference using Needleman-Wunsch with match score 1, mismatch cost −1, gap open cost −5, and gap extend cost 0. Reads with mean PHRED quality score below 30 were filtered. Mismatches at nucleotides with less than PHRED quality score 30 were filtered. Indels with less than three matching nucleotides on both sides with at least PHRED quality score 30 were filtered.

**Identifying peptide hits**. We excluded peptides with less than 100 reads in any experiment. We used a beta-binomial model to infer peptide editing effects from replicate data while adjusting for sampling noise from limited sequencing reads. We model a peptide $i$ with parameters $\alpha_i$, $\beta_i$ used to sample a peptide effect $p_{ij} \sim Beta(\alpha_i, \beta_i)$ for an experiment or batch $j$. Samples from experiment or batch $j$ are taken for sequencing, yielding a binomial distribution over the number of edited reads $y_{ij} \sim Bin(n_j, p_{ij})$ for read depth $n_j$. Given $k$ samples of $y_{ij}$ over $k$ biological replicates or batches, we infer the maximum likelihood estimate (MLE) of $\alpha_i$, $\beta_i$ for peptide $i$. As our beta-binomial model is conjugate, the MLE of $\alpha_i$, $\beta_i$ can be found analytically by solving the system of equations[27]: [Eqs.1–2]

$$0 = -k\Gamma(\alpha_i) - \Gamma(\alpha_i + \beta_i) + \sum_j \Gamma(y_{ij} + \alpha_i) - \Gamma(n_j + \alpha_i + \beta_i) \quad (1)$$

$$0 = -k\Gamma(\beta_i) - \Gamma(\alpha_i + \beta_i) + \sum_j \Gamma(n_j - y_{ij} + \alpha_i) - \Gamma(n_j + \alpha_i + \beta_i) \quad (2)$$

Where $\Gamma()$ is the Gamma function. We solved this using Sympy[1]. When solutions could not be found due to numerical instability, we used a fast approximation that solves the MLE of $\alpha_i$, $\beta_i$ by matching the observed mean and variance, motivated by viewing the beta-binomial distribution as an overdispersed binomial distribution.

The additional variance over a binomial distribution is related to the sum $\alpha_i + \beta_i$. [Eqs. 3–6]

$$obs\ var = var_j\left(y_{ij}/n_j\right) \qquad (3)$$

$$expvar = \frac{mean_j\left(y_{ij}/n_j\right) * \left(1 - mean_j\left(y_{ij}/n_j\right)\right)}{mean_j\left(n_j\right)} \qquad (4)$$

$$\alpha_i + \beta_i = \frac{mean\left(n_j\right) - 1}{(obsvar/expvar) - 1} - 1 \qquad (5)$$

$$\frac{\alpha_i}{\alpha_i + \beta_i} = mean_j\left(y_{ij}/n_j\right) \qquad (6)$$

To interpret $\alpha_i$, $\beta_i$, we convert them into the mean $\frac{\alpha_i}{\alpha_i + \beta_i}$ and variance $\frac{\alpha_i \beta_i}{(\alpha_i + \beta_i)^2(\alpha_i + \beta_i + 1)}$ of a Beta distribution.

We selected peptides for follow-up evaluation using several metrics. To increase confidence in hits, we preferred peptides present in higher numbers of replicates. We prioritized peptides based on the probability of observing a higher edited read count under its inferred peptide effect parameters compared to edited read counts sampled from inferred control peptide effect parameters under our beta-binomial model, which prefers higher MLE mean and lower MLE variance. We also selected peptides with high MLE mean effect even if their variance was high.

**100 target site library design**. An oligonucleotide pool containing 100 target sequences was synthesized by IDT. Each oligonucleotide contained the following elements 3′–5′: 19 nt PE1 stub, 4 nt barcode, ~40 nt variable target, 6 nt poly A terminator, ~30 nt PBS/template, 86 nt hairpin, 20 nt spacer, 20 nt U6 stub. The barcode stuffer allowed individual pegRNA and target sequence pairs to be identified after deep sequencing. To test the effect of PBS and RT template length on PE2 efficiency, we prepared pegRNAs with eight different combinations of edit types. Types of mutations:

· 3 × 1 nt substitution
○ PAM NGG– > NCG
○ PAM NGG– > NGT
○ Seed 1 transversion—nt nearest PAM, AàT, TàA, CàG, GàC
· 3 ×> 1 nt substitution
○ PAM NGG– > NCT
○ Seed 2–3 transversion (AàT, TàA, CàG, GàC) + PAM NGGàNTC [discontinuous, maintain 2 intervening nt]
○ 6 nt PAM + seed change to GAATTC
· 1 × 1 nt ins
○ PAM NGG– > NGTG
· 1 × 1 nt del
○ PAM NGG– > not NGG

delete 1st G unless G after PAM
Delete seed 1 unless 1st base of PAM is identical

To design a library of 100 pegRNA-target pairs we used 96 pegRNA-target pairs from Kim et al. high-throughput library data that vary in prime editing efficiency. In their library, for all sites with DeepSpCas9 score >20, average PE efficiency is 11%, SD = 9%. We chose 4 target sites 0–1 SD below average (1%, 3%, 5%, 8% PE), 4 sites around average (11%, 14%, 17%, 20%), 2 sites ~1 SD above average (30%, 40%), 2 near top (50%, 60%). Our library also includes 4 substitution mutations from Anzalone et al that showed highest prime editing activity. Oligo library was cloned into pLenti-sgRNA-FE vector using NEBuilder HiFi DNA Assembly Kit (NEB). Assembled plasmids were purified by isopropanol precipitation with GlycoBlue Coprecipitant (Thermo Fisher) and reconstituted in TE and transformed into Endura™ Electrocompetent Cells (Lucigen). After library diversity was verified, library mastermix was used to produce lentivirus.

**Production of lentivirus and cellular infection**. HEK293FT cells (15 × 10^6) were seeded on 150 mm cell culture dishes containing DMEM. The next day, cells were transfected with pCMV-VSV-G (Addgene #8454), pRSV-Rev (Addgene #12253), pMDLg/pRRE (Addgene #12251) and library, at a ratio of 1:2:3:4, using TransIT®-Lenti Transfection Reagent (Mirus Bio). At 8 h after transfection, cells were refreshed with maintaining medium. At 24 h and 48 h after transfection, the lentivirus-containing supernatant was collected, filtered through a 0.45 μm pore filter (Corning), concentrated using Lenti-X™ Concentrator (TakaraBio), aliquoted and stored at −80 °C.

In preparation for lentivirus transduction, cells (U2OS, HCT-116, mESC, HEK293FT) were seeded on 100 mm dishes (at a density of 2 × 10^5, 6.5 × 10^4, 6.5 × 10^4, 1 × 10^5 cells per cm^2) and concentrated lenti was added to the media. The cells were then incubated overnight, after which cells were refreshed with maintaining medium before adding blasticidin at 48 h and keeping it for

minimum of next 5 d to remove untransduced cells. To preserve its diversity, the cell library was maintained at a count of at least 1 × 10^7 cells throughout the study. Our initial 12k library, self-targeting screens, have been performed entirely in mESC via Tol2 stable integration. As a control, we used peptides of same length carrying sequence of house-keeping genes.

For the 100-target site library, cells were first transduced and selected for library integration, followed by transfection with PE2 editor vectors. Seven different mono, di, and tri peptide combinations of PE2 were tested, including the PE2 construct used to clone these peptide-PE2 constructs (CTRL-PE2, containing a short N-terminal linker) in generated cell lines carrying a 100-target library.

**Measurement of PE2 efficiencies at endogenous sites**. To validate the results of the high-throughput experiments, six individual pegRNA-encoding plasmids targeting endogenous NF2 and TP53 loci each were constructed and used to produce lentiviral particles. In preparation for transfection, HEK293T, A549, HCT116, and U2OS cells were seeded on 10 cm plates at a density of 4*10^4 cells per cm^2 and transduced with a lentivirus carrying pegRNA-encoding plasmid. After a week of selection for successful integration of constructs, cells were harvested for gDNA extraction followed by library preparation for NGS. Primers used to sequence NF2 and TP53 locus are listed in Supplementary Data 7.

**NLS prediction**. To predict potential NLS sequence, we used "cNLS mapper", available at https://nls-mapper.iab.keio.ac.jp/cgi-bin/NLS_Mapper_form.cgi. Using cut off value 5, recommended by creators of software.

**Reverse transcription quantitative PCR (RT-qPCR)**. Cell culture were then collected for purification or storage at −80 degrees. The precipitated mixture was then subjected to purification with RNeasy Mini Kit (QIAGEN), following the manufacturer's protocol. RNA was resuspended and quantified with Nanodrop (Thermo Fisher), and subjected to DNase treatment with RQ1 RNase-Free DNase (Promega), following the manufacturer's protocol. To generate cDNA from DNase-treated total RNA for samples to profile gene expression, we used 1–2 ug of DNase-treated RNA for each reverse transcription reaction with random hexamer from ProtoScript® First Strand cDNA Synthesis Kit (NEB); Afterward, cDNA was subjected to qPCR using Eva Green SYBR dye, and Q5 PCR mastermix (NEB) with CFX Connect Real-Time PCR machine (BioRad). For gene expression profiling, qPCR was performed with gene-specific qPCR primers. Ct readouts of each gene were first normalized with housekeeping gene GAPDH (ΔCt), and the relative expression of individual genes vs. the expression levels in control conditions was then calculated with 2-ΔΔCt method.

**Flow cytometry**. Cells were flow cytometrically sorted according to their GFP expression levels using BD Aria Sorter. Three populations of cells were collected: cells with top 20% GFP expression, next 20%, and remaining 60%. Cells were examined for GFP fluorescence using DxP11. Data were analyzed using FCS Express.

**Statistics**. Unless otherwise stated, results are presented as mean ± standard deviation. Statistical analyses of results were performed using GraphPad Prism 8 (for statistical details of each experiment, see figure legends). Statistically significant differences are denoted as follows: *$p < 0.05$, **$p < 0.01$, ***$p < 0.001$. For comparison of two independent groups, two-sided two-sample $t$-tests were used for normally distributed data with equal or similar variance (Student's $t$ test). A value of $p < 0.05$ was used to determine significance.

**Reporting summary**. Further information on research design is available in the Nature Research Reporting Summary linked to this article.

## Data availability

Plasmids encoding select cloning vectors have been deposited at Addgene for distribution. Custom code used to process and analyze peptide library data are available at https://github.com/maxwshen/prime-peptide. The NGS sequencing data generated in this study are provided in the Supplementary Information as well at https://dataverse.harvard.edu/dataset.xhtml?persistentId=doi:10.7910/DVN/WDOYRZ. Source data are provided with this paper.

## Code availability

Custom code used to process and analyze peptide library data are available at https://github.com/maxwshen/prime-peptide.

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

## Acknowledgements
The authors thank Mandana Arbab and Grigoriy Losyev for technical assistance. The authors acknowledge funding from 1R01HG008754 (R.I.S.), 1R21HG010391 (R.I.S., C.A.C.), American Cancer Society (R.I.S.), American Heart Association (R.I.S.), Qatar Biomedical Research Institute (R.I.S.), and the São Paulo Research Foundation- FAPESP n° 2019/13813-6 and 2017/25009-1 (L.C.Z.). Figures created with BioRender.

## Author contributions
Conceptualization, Methodology, Writing—Original Draft and Writing—Reviewing and Editing: R.I.S., M.V., L.L., M.W.S., J.D.F. Investigation and Validation: R.I.S., M.V., L.C.Z., M.C., E.A., D.B., M.W.S., J.D.F., T.Y.; Software, Formal Analysis and Visualization: R.I.S., M.W.S., J.D.F., M.V., T.Y.; Funding Acquisition and Supervision: R.I.S.

## Competing interests
The authors declare no competing interests.
