## [Peer Review File · Nature Communications]

Reviewers' Comments:

Reviewer #1:

Remarks to the Author:

The manuscript by Sherwood and colleagues describes the screening of a large population of peptides derived from DNA repair factors for sequences that increase the efficiency of Prime Editing in cell culture when fused to PE2. The authors further refine the peptides isolated from an initial large scale screen through screening a more focused sublibrary. They successfully identify a number of peptides that increase PE editing rates in both mESCs and HCT-166 cells. The top hits from the library are linked in tandem to afford higher rates of prime editing. The top candidate (IN-PE2) is evaluated on an endogenous target (NF2) at multiple sites in HEK293T and U2OS cells, where it modestly increases prime editing efficiency relative to the unmodified PE2.

Overall, this manuscript describes a novel approach to identify peptide sequences that improve prime editor function. The authors utilize multiple biological replicates in their screens to overcome inherently noisy data that is generated in the transposase-based screening system. In the context of the clever screening framework (assaying editing within the linker that joins the peptide library to PE2) they identify peptides that increase prime editing in multiple different cell types, and for the top peptide hits the independent impact on editing rates can be increased through multiplexing. Experiments to understand the mechanism of action for these peptides indicate that increased prime editor expression at the protein level could play a role in increased activity. Outside of the screening framework, the impact of the identified peptides on prime editor function at endogenous target sites is more moderate and variable.

Primary concern:

The modest and variable increase in prime editing activity at endogenous target sites for IN-PE2 relative to PE2. These data suggest that there is little advantage to utilizing IN-PE2 over PE2 at an endogenous target sequence. This reviewer wonders if the contrast between the editing efficiency in the library setting and the endogenous target sites is a function of the strong promoters that are utilized for the transgene cassettes that create an open chromatin environment? NF2 is expressed at higher levels in U2OS cells than HEK293T cells, so this conjecture may be incorrect. Nonetheless the authors should investigate editing rates at other genomic loci (is NF2 problematic for some reason) to provide evidence that incorporating the NI peptides provides utility for prime editing at endogenous targets.

Minor questions / issues:

- It is very interesting that a linear model is predictive for combined influence of the dual peptides on prime editing activity in the Pepseq framework. This suggests that the peptides are acting independently to increase the prime editing rate. However, analysis of many peptide combinations in the 100 member pegRNA target site library screen did not yield similar linear improvements in prime editing efficiency (Supplementary Figure 4B). Do the authors have thoughts about this dichotomy? Does this result suggest that some of the discovered peptide sequences are only effective at increasing prime editing activity in a subset of sequence/genomic contexts?
- In Supplementary Table 1 – there are two different peptides that are listed as peptide #1 for IGF1 – these peptides share similarities and differences in sequence. Do their relative rates of prime editing efficiency provide any insight into the sequence features that are useful for increasing prime editing efficiency for the more potent of these peptides?
- In Supplementary Table 2 the majority of the entries are not associated with gene names/peptide IDs, which limits the utility of this table to the reader since the peptide origin information is unknown.

Simple requests:

- Please include more information in the figure legend for Sup Fig 4A with regards to the meaning of the letters on the X-axis. It is unclear what these signify.

- It is unclear from the methods section how the sgPE-linker expression cassette was delivered subsequent to the Tol2 library cassette integration for the library screens.

Reviewer #2:

Remarks to the Author:

Prime editing is a promising precision genome editing technology with a high potential for many applications in genetics and medicine. Strategies that improve the efficiency of prime editing are critical to the advance of this important new technology.

In this paper, Velimirovic et al. fused small peptides (85 amino acids) to the N-terminus of the prime editor PE2 and measured the impact of the fusion proteins on prime editing efficiency. From their unbiased screen, the authors identified 105 prime editing enhancer peptides. Then, the authors tested whether combinatorial fusion of two or three enhancer peptides induces an additive effect on prime editing efficiency. These studies derived an IN-PE2 construct that they validated at six endogenous loci in two cell lines.

This paper has the potential to be an important addition to the growing list of strategies that increase prime editing efficiencies (Lin et al., *Nature Biotechnology* 2021, Liu et al., *Nature Communications* 2021, Song et al., *Nature Communications* 2021, Nelson et al., *Nature Biotechnology* 2021, Ferreira da Silva et al., *BioRxiv* 2021). However, the mechanism of action of the IN dipeptide is uncertain. The community might not democratize the use of IN-PE2 to increase prime editing without understanding how the dipeptide enhances prime editing efficiency in human cells.

While the rationale for fusing peptides derived from DNA repair proteins was attractive, the enhancer IN peptides has no role in DNA damage response (DDR). The only explanation proposed is that the dipeptide increases the amount of protein by potentially increasing transcription or translation, based on a duplicate FACS experiment. Experiments measuring the level of RNA could reinforce this possibility. Also, it would be expected that fusion of the IN dipeptide to the N-terminus of other proteins would have the same effect. An alternative hypothesis, not investigated by the authors, is that the dipeptide recruits particular cellular factor(s), potentially linked to the DDR, to stimulate prime editing lesion repair. A co-IP-MS using a commercial Cas9 antibody would be reasonable.

The authors do not discuss the origins of the peptides derived from NFATC2IP and IGF1. Are the peptides related to certain domains of these proteins? What are their cellular roles? The authors could also use "alpha fold" to predict if their peptides are expected to be structured, providing more insights into how the peptides stimulate prime editing. The authors also do not study the composition of these peptides. Do they contain NLS-like sequences that would increase PE2 localization in the nucleus, like observed by Liu et al. *Nature Communications* 2021?

Finally, the experiment that would provide solid proof that the role of the IN peptide is specific is by generating a point mutation into each peptide (I and N) and observe that the mutant suppresses the stimulatory effect of IN-PE2 compared to PE2. If these peptides are derived from domains of the proteins, it might be possible to introduce a point mutation that is known to disrupt their function. In line with this, it is surprising that fusion of a CDKN2A peptide to "NI"-PE2 suppresses the stimulatory effect of IN-PE2 (Supplementary Fig 4b) and that fusion of an MSH4 peptide to NFATC2IP suppresses the effect of the NFATC2IP peptide. The authors do not discuss these intriguing data, even though these results could explain the stimulatory effect induced by their dual and individual peptides.

Other comments:

- 1) Figures: There are a few issues with the main and supplementary figures. The x and y axes of Figs 1d, 1e, 1f, 2c, 2d, 2e, 2f, Supplementary figs 3 and 5 must start at 0 because they bias the interpretation of the data.
- 2) Fig 2f is a crucial figure of this paper as it shows that IN-PE2 stimulates prime editing efficiency

at endogenous sites. However, this figure has many problems: the y-axis does not start at 0. IN-PE2 is compared to CTRL-PE2 but not PE2. It is unclear why there are n = 10 and 12 because more data points are shown in the panel. The y-axis is labeled as "fraction", but it seems to refer to fold change.

3) In general, the legends of the figures have limited descriptions and details. No statistical analysis of the data is provided.

4) The authors compare the effect of IN-PE2 to CTRL-PE2 but not to PE2. How does CTRL-PE2 compare to PE2? They should compare the performance of IN-PE2 to PE2 because CTRL-PE2 might negatively impact PE2.

5) Why is "NI" used in Fig 2f and Supplementary Fig 6 and "IN" in others? The text also refers to "NI peptides" in line 155.

6) Line 102: "To gain insights into how these peptides function..." Dual fusion of peptides does not provide insights into how they stimulate prime editing. I would recommend that the authors remove this.

7) Recent strategies that improve prime editing are not discussed in the manuscript. The authors should put into context their work and link them to the recent work that also improved prime editing (Lin et al., Nature Biotechnology 2021, Liu et al., Nature Communications 2021, Song et al., Nature Communications 2021, Nelson et al., Nature Biotechnology 2021, Ferreira da Silva et al., BioRxiv 2021).

8) Fig 2g-h: It is unclear whether the addition of a GFP affects the stimulatory effect induced by IN-PE2. The authors should compare the effect to GFP-PE2.

Below, we have attached a detailed point-by-point response to the reviewers' comments. We have carefully addressed every reviewer comment through additional analyses, experiments, and clarifications to the text; we believe that these changes have improved the quality of our manuscript and further strengthened our findings. To summarize the major changes to the manuscript, we have:

- Expanded our comparison of IN-PE2 vs. PE2 at endogenous loci. We have now tested prime editing efficiency at 12 endogenous genomic loci in four cell lines, providing strong evidence that incorporating the IN peptides increases prime editing at endogenous targets. IN-PE2 treatment leads to significantly increased prime editing efficiency in all cell lines when considering all targeted loci (median 1.35X in A549, 1.92X in HCT-116, 1.17X in HEK293T, 1.78X in U2OS).
- Expanded our mechanistic analysis of the IN dipeptide. We have found that:
 - fusion of the IN dipeptide to the N-terminus of GFP increases protein level, suggesting that IN peptides are not specific to increasing levels of PE2.
 - the IN dipeptide increases prime editing efficiency when IN and PE2 are separated by a self-cleaving 2A peptide, suggesting that IN's primary function is independent of direct fusion with PE2.
 - the IN peptides do not increase transcription, suggesting that they are most likely to function through increasing translation.
- Updated our figures to provide more intuitive legends, higher-resolution panels, and more complete information on experiments.

In the section below, our responses are listed in blue.

Reviewer #1 (Remarks to the Author):

The manuscript by Sherwood and colleagues describes the screening of a large population of peptides derived from DNA repair factors for sequences that increase the efficiency of Prime Editing in cell culture when fused to PE2. The authors further refine the peptides isolated from an initial large scale screen through screening a more focused sublibrary. They successfully identify a number of peptides that increase PE editing rates in both mESCs and HCT-166 cells. The top hits from the library are linked in tandem to afford higher rates of prime editing. The top candidate (IN-PE2) is evaluated on an endogenous target (NF2) at multiple sites in HEK293T and U2OS cells, where it modestly increases prime editing efficiency relative to the unmodified PE2.

Overall, this manuscript describes a novel approach to identify peptide sequences that improve prime editor function. The authors utilize multiple biological replicates in their screens to overcome inherently noisy data that is generated in the transposase-based screening system. In the context of the clever screening framework (assaying editing within the linker that joins the peptide library to PE2) they identify peptides that increase prime editing in multiple different cell types, and for the top peptide hits the independent impact on editing rates can be increased through multiplexing. Experiments to understand the mechanism of action for these peptides indicate that increased prime editor expression at the protein level could play a role in increased activity. Outside of the screening framework, the impact of the identified peptides on prime editor function at endogenous target sites is more moderate and variable.

Response: We thank the reviewer for the positive feedback and their assertion of our manuscript's potential impact.

Primary concern:

1.1

The modest and variable increase in prime editing activity at endogenous target sites for IN-PE2 relative to PE2. These data suggest that there is little advantage to utilizing IN-PE2 over PE2 at an endogenous target sequence. This reviewer wonders if the contrast between the editing efficiency in the library setting and the endogenous target sites is a function of the strong promoters that are utilized for the transgene cassettes that create an open chromatin environment? NF2 is expressed at higher levels in U2OS cells than HEK293T cells, so this conjecture may be incorrect. Nonetheless the authors should investigate editing rates at other genomic loci (is NF2 problematic for some reason) to provide evidence that incorporating the NI peptides provides utility for prime editing at endogenous targets.

Response: Thanks for raising this important point about the importance of evaluating IN-PE2 at a broader set of endogenous loci. We have addressed this point thoroughly with new experimental data. We have expanded the number of endogenous test sites to 12 (6 in NF2 and 6 in TP53) and have now evaluated this collection of sites in 4 cell lines (HEK293T, U2OS,

A549, and HCT-116). This expanded dataset further confirms the advantage of incorporating the IN dipeptide for prime editing at endogenous targets.

We find that, across all targeted loci, IN-PE2 treatment yields significantly increased prime editing efficiency in all cell lines as compared to PE2 (median 1.35X in A549, 1.92X in HCT-116, 1.17X in HEK293T, 1.78X in U2OS, $p < 0.01$ in each cell line) (Figure 2f). While there is variability in the increase in prime editing at individual sites and across cell lines, these results indicate that IN-PE2 provides a robust increase in prime editing at endogenous loci.

To the question of the effects of the chromatin environment on prime editing, we do not believe our data explore a thorough enough collection of endogenous loci to draw conclusions on the interaction between PE2/IN-PE2 and the chromatin environment. We agree this is an important point for future investigation but believe it is out of the scope of the present study, as it would require exploring a substantial collection of endogenous sites with different chromatin environments to draw meaningful patterns in the presence of variable pegRNA efficiency.

Minor questions / issues:

1.2

- It is very interesting that a linear model is predictive for combined influence of the dual peptides on prime editing activity in the Pepseq framework. This suggests that the peptides are acting independently to increase the prime editing rate. However, analysis of many peptide combinations in the 100 member pegRNA target site library screen did not yield similar linear improvements in prime editing efficiency (Supplementary Figure 4B). Do the authors have thoughts about this dichotomy? Does this result suggest that some of the discovered peptide sequences are only effective at increasing prime editing activity in a subset of sequence/genomic contexts?

Response: We agree that some of the results from the 100-site library are confounding in light of the dual-peptide PepSeq data. To explore this point in more detail, we have provided a heat map of control-normalized mESC prime editing efficiency at the 13 sites within the 100-site library with $>0.5\%$ editing in control PE2-treated cells (Supplementary Fig. 7). We find that, by and large, peptides show consistent effects on prime editing across target sites. For example, IGF1-NFATC2IP-PE2 shows reproducibly increased editing at all 13 sites, IGF1-PE2 and NFATC2IP-PE2 show consistently more moderately increased prime editing. The three tripeptide prime editors consistently show no increase in prime editing across target sites. While we do not have evidence for this point, we believe that the failure of all of the tripeptide prime editors implies that the increased length of the tripeptides either sterically hinders PE2 or impairs translation efficiency and is thus not pertinent to the linear model results. Thus, our best answer to this question is that the single and dual peptides by and large consistently increase prime editing across target sites with magnitudes generally in line with the linear model, while the tripeptides fail for technical reasons. We have added a brief discussion to this end in the text.

1.3

- In Supplementary Table 1 – there are two different peptides that are listed as peptide #1 for IGF1 – these peptides share similarities and differences in sequence. Do their relative rates of prime editing efficiency provide any insight into the sequence features that are useful for increasing prime editing efficiency for the more potent of these peptides?

Response:

We have now systematically compared thousands of wt and phosphomimetic peptides from our 12,000-peptide library to see whether a wt peptide and its phosphomimetic counterpart are more correlated in fraction prime editing than two unrelated peptides. When taking the median prime editing rate from three replicates, phosphomimetic and wt peptides do not show any higher correlation in relative rates of prime editing efficiency than other pair of peptides ($R=0.03$) (Supplementary Figure 8). This may have to do with the overall noise in our 12,000-peptide data. However, given this general lack of correlation, we do not believe that comparing prime editing efficiency of IGF1p1 and IGF1pm1 in our 12,000-peptide dataset will shed mechanistic light on the function of IGF1pm1. Furthermore, IGFp1 and IGFpm1 differ at 20 positions, so it would require a more fine-grained mutation scanning paradigm to identify positions influencing the function of IGF1pm1.

1.4

- In Supplementary Table 2 the majority of the entries are not associated with gene names/peptide IDs, which limits the utility of this table to the reader since the peptide origin information is unknown.

Response:

We have revised Supplementary Table 2, to show the editing efficiency of each peptide from our library and clearly associate its name/ID. We note that certain peptides are not associated with gene names in the database we used to build this library, so these remain blank; however, most peptides are now annotated.

Simple requests:

1.5

- Please include more information in the figure legend for Sup Fig 4A with regards to the meaning of the letters on the X-axis. It is unclear what these signify.

Response: We have added additional information for our Supplementary Figure 4A, to include a more detailed description of the figure itself. Each point on X represents one of the 100 targets of the mini library whose sequence can be found in supplementary table 5.

1.6

- It is unclear from the methods section how the sgPE-linker expression cassette was delivered subsequent to the Tol2 library cassette integration for the library screens.

Response: For the 100-site library screen, we delivered the pegRNA/target site library to mESC, HCT-116, and U2OS cell lines by stable lentiviral integration (in line with Kim et al 2020), and then we subsequently transfected PE2/peptide-PE2 via lipofection followed by stable selection. We have updated our methods section to clearly distinguish 100-site library experiments from the initial 12k library, self-targeting screens that have been performed entirely in mESC via Tol2 stable integration.

Reviewer #2 (Remarks to the Author):

Prime editing is a promising precision genome editing technology with a high potential for many applications in genetics and medicine. Strategies that improve the efficiency of prime editing are critical to the advance of this important new technology.

In this paper, Velimirovic et al. fused small peptides (85 amino acids) to the N-terminus of the prime editor PE2 and measured the impact of the fusion proteins on prime editing efficiency. From their unbiased screen, the authors identified 105 prime editing enhancer peptides. Then, the authors tested whether combinatorial fusion of two or three enhancer peptides induces an additive effect on prime editing efficiency. These studies derived an IN-PE2 construct that they validated at six endogenous loci in two cell lines.

This paper has the potential to be an important addition to the growing list of strategies that increase prime editing efficiencies (Lin et al., Nature Biotechnology 2021, Liu et al., Nature Communications 2021, Song et al., Nature Communications 2021, Nelson et al., Nature Biotechnology 2021, Ferreira da Silva et al., BioRxiv 2021). However, the mechanism of action of the IN dipeptide is uncertain. The community might not democratize the use of IN-PE2 to increase prime editing without understanding how the dipeptide enhances prime editing efficiency in human cells.

Response: We thank the reviewer for the positive feedback.

While the rationale for fusing peptides derived from DNA repair proteins was attractive, the enhancer IN peptides has no role in DNA damage response (DDR). The only explanation proposed is that the dipeptide increases the amount of protein by potentially increasing transcription or translation, based on a duplicate FACS experiment.

2.1

Experiments measuring the level of RNA could reinforce this possibility.

Response: We thank the reviewer for highlighting the importance of determining the mechanism by which the identified peptides increase prime editing. We have performed several additional experiments and analyses that shed light on this issue.

First, we have performed more extensive targeting of endogenous loci in four human cell lines (see response to Reviewer 1), which support that IN-PE2 robustly increases prime editing in cells with proficient (A549, U2OS) and deficient (HCT-116, HEK293T) mismatch repair (Fig. 2f, Supplementary Fig. 9). This evidence further suggests that the IN peptides do not act via MMR. Second, we have cloned an IN-P2A-PE2 construct in which the IN peptides are separated from PE2 by a self-cleaving peptide and found that IN-P2A-PE2 increases prime editing to a similar degree as IN-PE2 (Supplementary Fig. 10). This experiment suggests that the IN peptides do

not require direct fusion with PE2 to function, indicating an indirect role such as increasing protein levels.

Third, we have fused the IN dipeptides to GFP and found robust (~2-fold) increase in cellular GFP protein levels (Supplementary Figure 11). This result suggests that the IN peptides increase the levels of other proteins beyond PE2 when fused.

Fourth, in order to address whether the IN dipeptides act through increased transcription or translation, we have performed RT-qPCR on stable CTRL-PE2-GFP and IN-PE2-GFP mESC cell lines. We find that CTRL-PE2-GFP has X-fold higher RNA expression in mESC than IN-PE2-GFP (Supplementary Fig. 11). This additional experiment suggests that the IN dipeptide does not increase transcription.

Altogether, the evidence we have collected best supports the conclusion that the IN peptides act by increasing cellular translation. We have added a discussion to this effect in the manuscript.

2.2

Also, it would be expected that fusion of the IN dipeptide to the N-terminus of other proteins would have the same effect.

Response: As mentioned in our response above, we have fused the IN dipeptides to GFP and found robust (X-fold) increase in cellular GFP protein levels (Supplementary Figure 11). This result suggests that the IN peptides increase the levels of other proteins beyond PE2 when fused, although we do not know how generalizable this paradigm is.

2.3

An alternative hypothesis, not investigated by the authors, is that the dipeptide recruits particular cellular factor(s), potentially linked to the DDR, to stimulate prime editing lesion repair. A co-IP-MS using a commercial Cas9 antibody would be reasonable.

Response: As described in our response 2.1, we believe that there is strong evidence that the IN peptides do not act specifically on DNA repair. In particular, we have cloned an IN-P2A-PE2 construct in which the IN peptides are separated from PE2 by a self-cleaving peptide and found that IN-P2A-PE2 increases prime editing to a similar degree as IN-PE2 (Supplementary Fig. 10). This experiment suggests that the IN peptides do not require direct fusion with PE2 to function, indicating that the IN peptides are unlikely to act in the DNA damage response, but rather increases translation of PE (Supplementary Figure 11)

2.4

The authors do not discuss the origins of the peptides derived from NFATC2IP and IGF1. Are the peptides related to certain domains of these proteins? What are their cellular roles? The authors could also use “alpha fold” to predict if their peptides are expected to be structured, providing more insights into how the peptides stimulate prime editing.

Response: NFATC2IPp1 (AA2-86) is in a disordered region with no prior annotated function. IGFpm1 (AA74-154) spans regions involved in IGF1 interaction with insulin receptor and IGF

receptors as well as a disordered region. As these IGF1 interactions occur extracellularly, protein annotations do not clarify how these peptides function to enhance prime editing. Altogether, none of these regions are known to impact DNA repair in a direct manner. Moreover, the nuclear localization signal (NLS) algorithm cNLS mapper does not predict either peptide to harbor an NLS. We have included text to this effect in the revised manuscript.

2.5

The authors also do not study the composition of these peptides. Do they contain NLS-like sequences that would increase PE2 localization in the nucleus, like observed by Liu et al. Nature Communications 2021?

Response:

As we note in response 2.4, the nuclear localization signal (NLS) algorithm cNLS mapper does not predict either peptide to harbor an NLS.

To explore this question further, we have cloned an IN-P2A-PE2 construct in which the IN peptides are separated from PE2 by a self-cleaving peptide and found that IN-P2A-PE2 increases prime editing to a similar degree as IN-PE2 (Supplementary Fig. 10). This experiment suggests that the IN peptides do not require direct fusion with PE2 to function. If the IN peptides were acting to alter PE2 localization, we would expect this to be dependent on direct fusion with PE2, and so we believe that the evidence disfavors altered cellular localization as a major role of the IN peptides.

2.6

Finally, the experiment that would provide solid proof that the role of the IN peptide is specific is by generating a point mutation into each peptide (I and N) and observe that the mutant suppresses the stimulatory effect of IN-PE2 compared to PE2. If these peptides are derived from domains of the proteins, it might be possible to introduce a point mutation that is known to disrupt their function.

Response:

We agree that scanning mutagenesis would be a fantastic additional set of experiments to explore the required residues in the IGF1pm1 and NFATC2IPp1 peptides; however, given that these peptides do not reside in obvious functional domains, this would be a laborious experiment to perform requiring hundreds of mutant peptides. We believe this to be beyond the scope of this manuscript which is aimed at identifying improved prime editors. We have mentioned this good idea in the discussion as a point of future inquiry.

2.7

In line with this, it is surprising that fusion of a CDKN2A peptide to “NI”-PE2 suppresses the stimulatory effect of IN-PE2 (Supplementary Fig 4b) and that fusion of an MSH4 peptide to NFATC2IP suppresses the effect of the NFATC2IP peptide. The authors do not discuss these intriguing data, even though these results could explain the stimulatory effect induced by their dual and individual peptides.

Response:

In the case of three tri-peptide prime editors, although we do not have evidence for this point, we believe that the failure of the tripeptide prime editors implies that the increased length of the tripeptides either sterically hinder PE2 or impair translation/transfection efficiency. Thus, our best answer to this question is that tripeptides fail for technical reasons and not as a result of specific mechanistic interference. We have added a brief discussion to this end in the text.

Other comments:

2.8

1) Figures: There are a few issues with the main and supplementary figures. The x and y axes of Figs 1d, 1e, 1f, 2c, 2d, 2e, 2f, Supplementary figs 3 and 5 must start at 0 because they bias the interpretation of the data.

Response:

We thank the reviewer for noting this. We have fixed this issue with new, corrected figures.

2.9

2) Fig 2f is a crucial figure of this paper as it shows that IN-PE2 stimulates prime editing efficiency at endogenous sites. However, this figure has many problems: the y-axis does not start at 0. IN-PE2 is compared to CTRL-PE2 but not PE2. It is unclear why there are n = 10 and 12 because more data points are shown in the panel. The y-axis is labeled as "fraction", but it seems to refer to fold change.

Response: We apologize, but this confusion is due to a lack of clarity on our end. In these endogenous locus experiments, the CTRL-PE2 refers to PE2 with a short 5-prime linker (5pl-PE2), which is the base construct used to clone peptide-PE2 constructs. We have clarified this terminology, as in earlier experiments in the manuscript we use CTRL-PE2 to refer to PE2 with control peptides attached to it (used in our initial, 12k screen where housekeeping genes are attached to PE2). We have also changed the features of the figure as you suggest.

2.10

3) In general, the legends of the figures have limited descriptions and details. No statistical analysis of the data is provided.

Response:

We have thoroughly revised all the figure legends and included necessary information.

2.11

4) The authors compare the effect of IN-PE2 to CTRL-PE2 but not to PE2. How does CTRL-PE2 compare to PE2? They should compare the performance of IN-PE2 to PE2 because CTRL-PE2 might negatively impact PE2.

Response : Please see our response to 2.9.

2.12

5) Why is “NI” used in Fig 2f and Supplementary Fig 6 and “IN” in others? The text also refers to “NI peptides” in line 155.

Response: We have corrected this typo.

2.13

6) Line 102: “To gain insights into how these peptides function...” Dual fusion of peptides does not provide insights into how they stimulate prime editing. I would recommend that the authors remove this.

Response: We thank the reviewer for the comment and removed this in the revised version of the manuscript.

2.14

7) Recent strategies that improve prime editing are not discussed in the manuscript. The authors should put into context their work and link them to the recent work that also improved prime editing (Lin et al., Nature Biotechnology 2021, Liu et al., Nature Communications 2021, Song et al., Nature Communications 2021, Nelson et al., Nature Biotechnology 2021, Ferreira da Silva et al., BioRxiv 2021).

Response:

At the time of preparation of this manuscript, the listed recent work had not been publicly available. We have mentioned and cited these relevant studies in the discussion section.

2.15

8) Fig 2g-h: It is unclear whether the addition of a GFP affects the stimulatory effect induced by IN-PE2. The authors should compare the effect to GFP-PE2.

Response:

The objective of this experiment was to ask whether the IN peptides increase the levels of PE2 in cells, and thus we did not assess the editing efficiency of IN-GFP-PE2. We do not advocate the use of IN-GFP-PE2 for any functional prime editing experiments, and we have now also shown that the IN peptides increase the expression of GFP to a similar degree to their increase in expression of GFP-PE2.

REVIEWER COMMENTS

Reviewer #1 (Remarks to the Author):

My apologies to the authors for the delay in returning this review. COVID made the last few weeks a challenge.

The revised manuscript by Sherwood and colleagues adds data to their original manuscript that provides a more thorough evaluation of IN-PE2 editing rates at alternate endogenous target sites (p53) in a variety of different cell lines. The authors also explore more deeply the mechanism of IN peptide function with regards to achieving increased editing rates across the assayed target sites, which reveals that IN peptide function may be operating at the level of increased PE protein levels through increased transcription and/or translation. Overall their results highlight the utility of PepSeq for the identification of peptide sequences that have improved function in the context of prime editing systems.

Response: We thank the reviewer for the positive feedback and their assertion of our manuscript's potential impact.

In their revised manuscript the authors have addressed the majority of my minor concerns.

Remaining concerns:

1) The authors provide additional analysis of the impact of their IN-PE2 system on the editing rates at a broader set of endogenous target sites and in a broader range of cell lines. While the median rates of prime editing are improved across the target sites as a group, there is variation in the editing rates for the control relative to the IN-PE2 across the different target sites. For example, the E285>H285 P53 site shows about 10 fold improvement in editing rate, while the G302>D302 P53 site shows a 3 to 5 fold reduction in the editing rate [sup fig 8]. Thus, the additional target sites added in the revision display continued variability in activity. Despite this (unless this reviewer is misinterpreting the data) the authors describe the results as

“indicate that IN-PE2 provides a robust increase in prime editing at endogenous loci”

The term “robust increase” could be softened to more accurately reflect the variability in the data given that in some cases the peptide fusion appears to decrease editing efficiency.

Response: We agree that prime editing remains variable across endogenous target sites in the presence and absence of IN-PE2. We note that other improved prime editing tools such as pegRNA stabilization and inhibition of mismatch repair also show a high degree of site-to-site variability in whether and by how much they improve editing (Nelson, J.W., et al, Nat Biotechnol (2022), Supplementary Figure 4 and 14; Liu, B., Dong, X., Cheng, H. et al. Nat Biotechnol (2022), Extended Data Fig. 2e). We have rewritten this part of the manuscript to more accurately reflect the data of peptide fusion editing efficiency, and we have noted that other recent work also shows variability in improvement of editing efficiency across target sites.

2) In Figure panel 2F – the authors indicate that the log of the ratio IN-PE2 to Control-PE2 editing rates is 100-fold at some sites in some cell lines. This is confusing. Is the plotted data the log of a ratio? It is unclear to this reviewer how this can be producing values ≥ 100 . [My apologies if I am missing something]

Response: We thank the reviewer for pointing out this error in the axis label. We plot the ratio of editing rate, not the log of this ratio (the axis is logarithmic, but not the ratio). We have corrected this in figure panel 2F.

Other minor issues:

The supplementary figures contain some discrepancies and are missing some panels based on the supplementary figure legends. In addition, the legends in this manuscript remain underdeveloped. This reviewer finds the contents of some figure panels confusing (e.g. fig 2f, as noted above) and the short legends lack necessary detail to interpret the figures.

Response: We thank the reviewer for pointing out this deficiency of our manuscript. We have improved figure legends to contain more information and allow for easier interpretation.

a) The Supplementary figure 4 legend indicates that there are three figure panels, but only one is present.

Response: During uploading of our manuscript, Supplementary Figure 4 was divided into 2 pages, which resulted in a misleading figure legend. We have corrected this issue.

b) In supplementary figure 8 – the top All cell line figure panel indicates no values >10 fold but in the subsequent panels in this figure many of the sites have values approaching 100 fold difference. This discrepancy indicates that one of the panels is incorrectly plotted.

Response: We have corrected the scale in Supplementary Figure 8.

c) The Supplementary 8 Figure legend indicates HEK293T cell data is included in these panels, but this cell line is not indicated in the plots.

Response:

We have corrected this error.

Reviewer #2 (Remarks to the Author):

In this revised manuscript, Velimirovic et al. added more experiments to propose that IN-PE2 increases the efficiency of prime editing through an increase in translation. The authors tagged IN-PE2 with GFP and measured the abundance of GFP signals by flow cytometry to demonstrate an increase in protein levels. They have also tested IN-PE2 in additional endogenous sites and cell lines. Unfortunately, I do not find that the new experiments have strengthened their conclusions. Moreover, the requested changes in the figures and figure legends have not been adequately addressed. Overall, I do not find that the revised manuscript is suitable for publication in Nature Communications, especially compared to recent papers published in the same journal (Liu et al., Song et al., and Ferreira da Silva et al.).

Response: We thank the reviewer for the constructive feedback.

Here are some comments for the authors.

1) The GFP experiments and quantifications are not described in the material and methods section, making it difficult to evaluate the quality of the experiments and determine the significance of the results. Nevertheless, the conclusion that IN-PE increases the abundance of PE2 by 1.5-fold appears incorrect. From FACS experiments, the only conclusion that can be drawn is that a certain fraction of the cells expressing IN-PE2 has a higher GFP signal compared to the cells expressing PE. Looking at the FACS profiles, it looks like ~5% of the cells have an increased GFP signal, while most of the cells have overlapping GFP signals. I am not convinced that this is significant and is responsible for the stimulation of PE. In addition, the link between the increase in translation and stimulation of prime editing has not been established. The authors could easily demonstrate their model by sorting IN-PE2-expressing cells with superior GFP signals and comparing the editing frequency to GFP+ PE2-expressing cells.

Response:

To address the link between increased IN-PE2 and increased prime editing, we have sorted pegRNA-treated cells that constitutively express IN-GFP-PE2 into three populations based on GFP fluorescence. We find increased levels of prime editing in sorted populations that correlate with their increased levels of GFP fluorescence (Supplementary figure 9d). These data demonstrate a link between increased IN-GFP-PE2 levels and prime editing stimulation, and with our other data strengthen our argument that increased IN-PE2 translation is responsible for increased prime editing. We thank the reviewer for suggesting this enlightening experiment.

Regarding interpretation of the GFP flow cytometry experiments, we have added a description of them in the materials and methods section. It is our interpretation that in both the flow cytometric comparison of IN-GFP-PE2 vs. GFP-PE2 and IN-GFP vs. GFP, there is a population shift in GFP, not merely a small subpopulation with increased fluorescence (Supplementary figure 9, Supplementary table 8). We have added a quantification of these flow cytometry plots where we calculate the percentage of cells above several distinct fluorescence thresholds, which supports the conclusion that samples with IN peptide attachment do indeed have a population shift in GFP levels. We have added clarification of this point to the manuscript (Supplementary table 8).

2) The editing data are presented as a fold-change rather than direct prime editing frequency throughout the manuscript. It is critical to show actual editing frequency because a few sites with a low editing frequency can artificially increase the overall fold-change. It is critical to provide a panel in main figures showing the raw percentage of editing between PE2 and IN-PE2 to give the reader direct access to this important information. Moreover, a supplementary table containing all the editing frequencies at each endogenous site, rather than fold-change, is a standard practice in the field.

Response:

We have included this data in the revised manuscript version in the form of a supplementary table and supplementary figures. Supplementary table 7 now contains prime editing frequency for each of the endogenous edits from Figure 2f, as well as the control normalized values, and raw data. This is in line with other papers reporting prime editing improvements (Nelson et al, 2021; Anzalone et al 2021; Ferreira da Silva et al 2022). We have also added a set of supplementary figures reporting percent editing at each site in each cell line tested (Supplementary Figure 7 and 8, revised). The exponential differences in prime editing efficiency between sites make a compiled figure with editing percentage difficult to interpret, and so we have chosen not to include all of these plots in the main figure panels.

3) The authors should remove the mention of “peptides derived from human DNA repair-related proteins” in the abstract as it misleads the reader that the identified peptides are affecting DNA repair.

Response: We agree that mentioning the origin of the peptides in the abstract may be misleading. We have removed this from the abstract.

4) The problems with the figures and figure legends have not been addressed: The authors have not corrected the axes as requested (Fig. 1e and 2c). Fig. 2f, which is one of the most important figures of their manuscript, also has issues. Why do the authors now use “Log fraction edited” instead of a “fraction of edited” like intended before? The authors should present the absolute percentage of editing from PE2 and IN-PE, not only as a fold change compared to 1 (for the reasons explained above, see comment 2).

Response: We have corrected the mistakenly log labeled axis in Figure panel 2F. We thank the reviewer for bringing this to our attention. Figure 2f shows fraction editing rates of IN-PE2 at endogenous loci normalized to CTRL-PE2. We have also added a set of supplementary figures reporting percent editing at each site in each cell line tested (Supplementary Figure 7 and 8, revised) as well as a Supplementary Table 7 with prime editing frequency for each of the endogenous edits from Figure 2f, as well as the control normalized values, and raw data.

5) The request to clarify the figure legends has not been satisfactorily addressed. Overall, the figure legends are too limited to understand what has been done and interpret the figure panels. For example, “Comparison of prime edited fraction for IN-PE2 vs. CTRL-PE2 at a set of twelve endogenous sites in HEK293T, HCT116, A549, and U2OS.” These changes are insufficient to understand the figure: What does n mean? What are the two points for each locus? Why does

the legend mention twelve endogenous sites when I find only ten annotated in the figure (on the right side)? The authors should improve the figure legends to help the readers understand their experiments and presented data.

Response: We thank the reviewer for bringing this to our attention. We have added more necessary details to figure legends.

6) The discussion on a very trending topic is minimal.

Response:

Our revised version of the manuscript has an extended discussion section.